# The identity blueprint: Decoding the professional identity of Basic Sciences Medical Teachers in Pakistan by developing and pilot testing a questionnaire

Faiza Kiran[ID][1]*, Nadia Shabnam[2]*, Shazia Irum[3], Samreen Misbah[ID][4], Asiya Zahoor[5], Rukhsana Ayub[6]

1 Associate Professor, Shifa School of Health Professions Education Shifa Tameer e Millat University, Islamabad, Pakistan, 2 Assistant Professor, Department of Health Professions Education, National University of Medical Sciences, Rawalpindi, Pakistan, 3 Assistant professor, Department of Health Professions Education, Shifa College of Medicine, Islamabad, Pakistan, 4 Associate professor Department of Community Medicine, Army Medical College, National University of Medical Sciences, Rawalpindi, Pakistan, 5 Assistant professor, Department of Health Professions Education National University of Medical Sciences, Rawalpindi, Pakistan, 6 Assistant Dean Assessment NUMS Department of Health Professions Education National University of Medical Sciences, Rawalpindi, Pakistan

* drfaizakiran@gmail.com, faiza.kiran.scm@stmu.edu.pk (FK); nadia.shabnam@numspak.edu.pk (NS)

## Abstract

### Introduction

The goal of the current study was to identify factors forming the professional identity of medical teachers teaching basic health sciences by developing and validating a questionnaire.

### Methodology

A self-administered questionnaire was developed with items on community of practice, passion for teaching, students' feedback, work-life balance, religious values, administrative support, work environment, job satisfaction, societal apathy, and opportunities for professional growth. The questionnaire was created after conducting interviews with faculty of basic health sciences, pilot tested, modified and verified by expert panel. Overlapping items and those which did not represent themes were removed and a final 20-item questionnaire was created. Finally, exploratory factor analysis was performed. The sample size was 301, and the participants recruited, regardless of gender, were medical teachers teaching basic health sciences, and had teaching experience of more than one year.

### Results

A total of 20 high-loading components made up the five variables of Exploratory Factor Analysis, which explained 50% of the variance. The 20-item questionnaire was

**Data availability statement:** All relevant data are within the paper and its Supporting Information files.

**Funding:** The author(s) received no specific funding for this work.

**Competing interests:** The authors have declared that no competing interests exist.

found to have satisfactory psychometric qualities. Five major constructs grouped as "CLaSiC-R" were unveiled and termed as identity blueprint of basic sciences medical teachers. They are Community of practice, Legacy, job Satisfaction, Commitment to excellence and Resilience. Cronbach's alpha of the 20 items was 0.648.

## Conclusion

The CLaSiC-R determines the five major constructs that form the professional identity of basic sciences medical teachers. The scale serves as a valuable indicator of key areas of professional development namely, socialization, religious values, emotional intelligence, motivation, job roles, committment to excellence, and work environment. Reforms in service structure and structured faculty development programs might facilitate their professional journey.

## Introduction

Professional identity formation (PIF) is a multifaceted and dynamic process that is fundamental to the development of competent, reflective, and resilient medical professionals. It involves the integration of personal values with professional roles through a process shaped by cultural norms, mentorship, self-reflection, experiential learning and most importantly socialization, within communities of practice [1,2]. While this process has been extensively explored in relation to medical students and clinicians, the professional identity of medical teachers especially those in basic medical sciences—remains significantly underexplored [3].

Medical teachers are central to the quality and effectiveness of medical education. They wear multiple hats as teachers, researchers, mentors, and administrators, within challenging environments [4]. Their professional identity—the way they perceive themselves in their teaching roles—profoundly influences their career satisfaction, motivation, engagement with students, and willingness to grow professionally [5,6]. Yet, in Pakistan, where sociocultural expectations and gender roles deeply influence career trajectories, the development of professional identity among medical teachers is still not well understood [7].

The Pakistani medical education landscape has undergone rapid transformation in the past two decades, with a surge in the number of private medical institutions [8,9]. Despite this growth, there is a lack of empirical research on how faculty, particularly in basic sciences, navigate their professional identities within these evolving academic environments. Cultural expectations, institutional dynamics, gender roles, and limited faculty development opportunities all contribute uniquely to the identity formation process in this context [5].

The identity of a basic sciences medical teacher (BSMT) often differs from that of clinical teachers due to the nature of their work, limited patient interaction, and unique academic responsibilities. This difference is compounded by a lack of structured socialization and mentorship opportunities, which are critical to foster a strong professional identity [10 ,11].

Previous studies have extracted factors affecting professional identity of medical students, trainees, physicians and basic sciences medical teachers, by qualitative analysis. Only fewer studies have approached the problem with quantifiable means; an essential measure to quantify attitudes, values and beliefs of physicians, trainees, and medical teachers. Of these a 10-item Professional Identity Questionnaire (PIQ) [12] and a 9-item Professional Self Identity Questionnaire (PSIQ) [13] in 2021, were developed for medical students. Another one, in 2015, was developed to quantify identity transformation in medical students, starting from premedical era till graduation. It has 10 key aspects, 06 domains and 30 subdomains [14]. Only one questionnaire by Tagawa in 2019, a 15-item scale, was found which measures PIF in residents, experienced instructors and medical students [15].

In 2017, a review on teacher identity, in the university context, identified 59 studies; 57 of whom applied qualitative methods, such as interviews or focus group discussions to assess teacher identity. Qualitative assessments of teacher identity, though gives broader scope, but fails to provide an economical application to repeated measurements or large-scale assessments [16], for example when questioning medical teachers in entire hospital or medical college. In 2006, an instrument measuring physicians' teacher identity was developed by Starr et al., including 37 items nested in nine scales designed from previously conducted interviews [17].

Though an important area, not a single study has been found where professional identity of basic sciences medical teachers has been measured by quantitative means. Understanding the specific factors that shape the PIF of BSMT is crucial not only for supporting individual faculty development but also for improving the overall quality of medical education. A deeper insight into this process can inform institutional policies, faculty development programs, and mentorship models tailored to local sociocultural realities.

Therefore, this study aims to identify and quantify the key factors contributing to the professional identity formation of basic sciences medical teachers, an area of research that remains largely overlooked but is essential for the advancement of medical education.

## Methodology

It was an exploratory sequential, mixed method study design where in the quantitative portion was conducted after the qualitative phase. Initially, a qualitative study was conducted by text-based interviews on Google forms, sent via emails to the basic medical sciences faculty followed by iterative data analysis [9]. Based on findings, a 20-item questionnaire was formed (See S1 Appendix at the end of manuscript). It was pilot tested on 5 participants. Questionnaire was uploaded on Google forms, distributed via social media to basic sciences medical teachers in Pakistan.

### Sampling and participants

The sample size for this survey-based study was determined based on methodological recommendations for factor analysis, which suggest a participant-to-item ratio of at least 10:1. As our questionnaire consisted of 20 items, a minimum of 200 participants was required. We recruited 301 participants, which meets this criterion and ensures sufficient statistical power for meaningful analysis. This approach aligns with recommendations in existing literature for ensuring adequate sample size in survey-based research used for factor analysis. Convenient sampling was done, and data was collected from basic sciences medical teachers of the country, by distributing questionnaires via emails, and WhatsApp groups. The survey was designed as a self-administered and self-completed online questionnaire. As compared to face-to-face or phone methods, distributing scales online removes interviewers, and ensures anonymity and confidentiality. Responses were taken on a Likert scale which makes it more feasible for the respondents to answer. In addition, randomized response format was ensured by using indirect or reverse questioning, which tend to reduce common methods and social desirability bias [18].

A total of 301 teachers teaching basic medical sciences provided their perceptions on factors forming their professional identity by responding to questionnaire. A total of 228(76%) females and 73(24%) males responded to the questionnaire

(Table 1). Of these, only 58(19%) were PhDs, 185(62%) were Masters and 58(19%) were medical/dental graduates (Table 1). Participants' official language and medium of instruction in medical institutions is English.

## Questionnaire development and validation

The questionnaire was developed and validated in 3 phases [19]. Questionnaire development was done in the first two phases while the last phase covered the questionnaire validation.

**Phase 1-Item development.** Questionnaire, in English, was developed after personal observations, and in-depth interviews of forty volunteer faculty members [9]. Initially, the questionnaire had 22 items, all about factors forming the professional identity of basic medical sciences teachers, to be rated on a 5-point Likert scale ranging from 1: strongly disagree to 5: strongly agree.

Items were identified through strategies mentioned earlier and represented all identified factors forming professional identities of medical teachers including community of practice, passion for teaching, students' feedback, work-life balance, religious values, administrative support, work environment, job satisfaction, societal apathy, and opportunities for professional growth [9]. Pilot testing of the questionnaire was done on five medical teachers. They were briefed about the objectives of the study. Revamping of items was done following their feedback. Items were phrased to avoid vagueness and biases, negatives, and double-barreled questions. These steps were taken to ensure content validity [19].

**Phase 2-Scale development.** After revising the questionnaire, medical teachers were approached again, deliberation was done till no additional issues were identified. In addition, expert panel judgment was engaged to ensure item clarity.

Table 1. Frequency and percentages of Faculty of different gender and qualifications, agreeing with the statements.

| No | Items | Gender | | Qualification | | | |
|---|---|---|---|---|---|---|---|
| | | Male 73(100) | Female 228(100) | MBBS 50(100) | BDS 8(100) | M Phil/ MS 185(100) | PhD 58(100) |
| 1 | I wish I could rewind the time and chose clinical field | 21(29) | 65(28) | 12(24) | 4(50) | 54(29) | 16(27) |
| 2 | I feel proud to be a medical teacher | 66(90) | 213(93) | 46(92) | 8(100) | 171(92) | 54(93) |
| 3 | It syncs well with family life | 67(92) | 214(94) | 45(90) | 8(100) | 173(93) | 55(95) |
| 4 | My seniors encourage me grow professionally | 50(68) | 166(73) | 35(70) | 5(62) | 138(74) | 38(65) |
| 5 | I wish I could practice as a part-time physician | 50(68) | 136(60) | 34(68) | 5(62) | 114(62) | 33(57) |
| 6 | My hard work is usually acknowledged at workplace | 47(64) | 161(71) | 32(64) | 5(62) | 131(71) | 40(69) |
| 7 | I have started accepting people the way they are | 61(83) | 201(88) | 40(80) | 6(75) | 166(90) | 50(86) |
| 8 | I stay calm in stressful situations | 55(75) | 173(76) | 35(70) | 5(62) | 145(78) | 43(74) |
| 9 | I take my students' words as personal evaluation of myself | 60(82) | 192(84) | 41(82) | 7(87) | 153(83) | 51(88) |
| 10 | I keep myself updated with the latest trends in my field | 64(88) | 207(91) | 43(86) | 8(100) | 165(89) | 55(95) |
| 11 | I foresee myself in a better position in my career | 62(85) | 177(78) | 39(78) | 5(62) | 148(80) | 47(81) |
| 12 | I am glad I bring positive changes in society | 57(78) | 135(59) | 41(82) | 8(100) | 159(86) | 47(81) |
| 13 | The only reward I get at workplace is, more work! | 43(59) | 133(58) | 30(60) | 5(62) | 112(60) | 29(50) |
| 14 | I am positively challenged at workplace | 52(71) | 167(73) | 36(72) | 7(87) | 137(74) | 39(67) |
| 15 | Teaching comes naturally to me | 53(73) | 197(86) | 34(68) | 6(75) | 159(86) | 51(88) |
| 16 | I feel accomplished when I see my students progressing | 70(96) | 222(97) | 47(94) | 8(100) | 181(98) | 56(96) |
| 17 | People think medical teachers are not doctors | 39(53) | 179(78) | 33(66) | 4(50) | 143(77) | 38(73) |
| 18 | I consider my profession as a legacy (sadqaejaariah) | 60(82) | 206(90) | 44(88) | 8(100) | 166(90) | 48(82) |
| 19 | I enjoy liberty to exercise innovative ideas at workplace | 56(77) | 177(78) | 34(68) | 7(87) | 147(79) | 45(24) |
| 20 | I frequently attend workshops, conferences, seminars etc. | 55(78) | 195(85) | 40(80) | 7(87) | 154(83) | 49(84) |

These steps were taken to obtain response process evidence [20,21]. The expert panel included health professions teachers with minimum qualifications of a master's degree in health professions education and a minimum of 03 years of professional experience. According to their feedback, the original version's 22 items were reduced to 20 items, and the questionnaire was revised to ensure translational validity (face validity) [12,13]. The questionnaire was developed and designed in the English language.

The final version of the scale consisted of 20-items. These items were evaluated using a Likert scale with a scoring range between 1 (strongly disagree) and 5 (strongly agree).

The drafted survey questionnaire was uploaded on Google forms and distributed via WhatsAppand email to basic medical sciences faculty. A total of 301 medical teachers, teaching basic medical sciences, regardless of gender, and had teaching experience of more than 01 year filled the questionnaire. Approval was obtained from Ethical Committee of, National University of Medical Sciences, Pakistan on 30 Jan 23 (#06/IRB&EC/NUMS/01/10682).

Participation was voluntary, informed consent was taken, and data was collected anonymously. The distribution and completion of the questionnaires took 06 months; from 25 February 2023 to 25 August 2023. To minimize response bias and social desirability bias, the survey was administered anonymously using an online platform. Participants were informed that their responses would remain confidential and that no identifying information would be collected. They were also assured that the data would be used solely for research purposes. These measures were taken to promote honest and unbiased responses.

**Phase 3-Scale evaluation.** In this phase, the strength of each item in a questionnaire is tested by determining the factor structure of the instrument, using exploratory factor analysis [22,23]. The descriptive statistics of all items including averages and standard deviations, as well as bivariate correlations, were also acquired. The statistical program SPSS 27.0 was used to analyze the data. Prior to EFA, Kaiser–Meyer–Olkin (KMO) measure of sample adequacy and Bartlett's test of sphericity were applied to check the minimum standard for the application of EFA. The instrument's internal consistency was determined by measuring Cronbach's alpha coefficient [24].

## Analysis and results

The participants from different regions of Pakistan responded to questionnaire, yielded 5 constructs and 20 items. The analysis was done in two parts. In the first part, we explored the data in the form of frequencies and percentages of the study items according to gender and qualification of medical teachers (Table 1) as well as their designations and years of experience (Table 2). Similarly, the reliability of the scale and item wise reliability was also obtained. The result showed that all 20 items are related to Cronbach Alpha (0.648).

Table 1 shows that male teachers 57(78%) self-acknowledge their positive societal contribution, more than female teachers 135(59%); item 12, "I am glad I bring positive changes in society" aligns with the above claim. They 62(85%) foresee better prospects in their career than females 177(78%); Item 11, "I foresee myself in a better position in my career" supports this statement. In addition, males have more desire to practice in the clinical field 50(68%) as compared to females 136(60%) probably because they are still the main breadwinners of the family in a society like Pakistan; item 5, I wish I could practice as a part-time physician", reinforces the above claim.

In contrast, though more females 197(86%) acknowledge their innate abilities and a natural predisposition towards teaching as compared to males 53(73%); Item 15, "Teaching comes naturally to me" aligns with this. At the same time, females 179(78%) seem more affected by societal apathy as compared to males 39(53%); item 17, "People think medical teachers are not doctors" agrees with this.

Table 2 results show that lecturers 27(32%) and assistant professors 24(27%) and those who have less than 5 years of experience 42(28%), show dissatisfaction and a desire to be a clinician rather than a teacher; item 1, "I wish I could rewind the time and chose clinical field" aligns with the above claim. However, with increasing experience in the field, and promotion to senior designation, fewer teachers show dissatisfaction.

**Table 2.  Frequency and percentages of Faculty, at different designations and years of experience, agreeing with the statements.**

| No | Items | Designation | | | | Years of Teaching Experience | | | |
|----|-------|-------------|--|--|--|------------------------------|--|--|--|
| | | Lecturer | Assistant Professor | Associate Professor | Professor | 1-5 | 6-15 | 16-20 | >20 |
| | | 85(100) | 90(100) | 59(100) | 67(100) | 149(100) | 70(100) | 38(100) | 44(100) |
| | | N(%) | N(%) | N(%) | N(%) | N(%) | N(%) | N(%) | N(%) |
| 1 | I wish I could rewind the time and chose clinical field | 27(32) | 24(27) | 19(32) | 16(24) | 42(28) | 24(34) | 9(24) | 4(9) |
| 2 | I feel proud to be a medical teacher | 78(92) | 81(90) | 56(95) | 64(95) | 137(92) | 65(93) | 36(95) | 41(93) |
| 3 | It syncs well with family life | 76(89) | 83(92) | 59(100) | 63(94) | 134(90) | 70(100) | 37(97) | 40(90) |
| 4 | My seniors encourage me grow professionally | 59(69) | 61(68) | 44(74) | 52(77) | 102(68) | 51(73) | 24(63) | 39(88) |
| 5 | I wish I could practice as a part-time physician | 59(69) | 50(55) | 34(57) | 43(64) | 96(64) | 44(63) | 20(52) | 26(59) |
| 6 | My hard work is usually acknowledged at workplace | 56(66) | 55(61) | 42(71) | 55(82) | 96(64) | 46(66) | 25(66) | 41(93) |
| 7 | I have started accepting people the way they are | 70(82) | 80(89) | 51(64) | 61(91) | 127(85) | 63(90) | 32(84) | 40(90) |
| 8 | I stay calm in stressful situations | 60(70) | 73(81) | 44(74) | 51(76) | 108(72) | 54(77) | 34(89) | 32(72) |
| 9 | I take my students' words as personal evaluation of myself | 70(82) | 72(80) | 55(93) | 55(82) | 120(80) | 62(88) | 34(89) | 36(81) |
| 10 | I keep myself updated with the latest trends in my field | 71(83) | 80(89) | 55(93) | 65(97) | 130(87) | 65(93) | 34(89) | 42(95) |
| 11 | I foresee myself in a better position in my career | 67(79) | 72(80) | 46(78) | 54(80) | 118(79) | 55(78) | 31(81) | 35(79) |
| 12 | I am glad I bring positive changes in society | 74(87) | 73(81) | 50(85) | 58(86) | 128(86) | 60(86) | 30(79) | 37(84) |
| 13 | The only reward I get at the workplace is more work! | 54(63) | 52(58) | 31(52) | 39(58) | 94(63) | 32(46) | 24(63) | 26(59) |
| 14 | I am positively challenged at workplace | 57(67) | 65(72) | 41(69) | 56(83) | 96(64) | 55(78) | 31(81) | 37(84) |
| 15 | Teaching comes naturally to me | 68(80) | 69(77) | 55(93) | 58(86) | 119(80) | 60(88) | 32(84) | 39(88) |
| 16 | I feel accomplished when I see my students progressing | 80(94) | 87(97) | 59(100) | 66(98) | 143(96) | 68(97) | 37(97) | 44(100) |
| 17 | People think medical teachers are not doctors | 69(81) | 65(72) | 39(66) | 45(67) | 116(78) | 51(73) | 24(63) | 27(61) |
| 18 | I consider my profession a legacy (Sadqa jaariah) | 76(89) | 76(94) | 56(95) | 58(86) | 130(87) | 61(87) | 35(92) | 40(90) |
| 19 | I enjoy liberty to exercise innovative ideas at workplace | 62(72) | 66(73) | 47(80) | 58(86) | 104(70) | 59(84) | 30(79) | 40(90) |
| 20 | I frequently attend workshops, conferences, seminars etc. | 70(82) | 69(77) | 50(85) | 61(91) | 117(78) | 58(83) | 34(89) | 41(93) |

Overall, both in **Table 1** and **Table 2**, factors like passion, self-regulation, work life balance, students' feedback, professional colleagues and society, desire to leave a legacy all were found constant and high in all teachers regardless of gender, designation, years of experience and postgraduate qualifications.

The result of items used for developing scale along with mean and standard deviation of scores are given in **Table 3**. Among the 20 items analyzed, basic sciences medical teachers demonstrated the highest mean scores for 9 items. The mean and standard deviation values reveal that basic sciences medical teachers exhibit a mixed pattern of professional identity attributes. Higher mean scores on four reverse-coded items reflect regret, low satisfaction, weak professional identity, negative self-perceptions, so higher agreement on these items indicates weaker professional identity whereas relatively lower means on items related to workplace acknowledgment, work–life integration, and institutional support suggest that participants feel under-recognized, unsupported, or strained in balancing professional and personal life which increases the risk of burnout. They might be experiencing insufficient structural, administrative, or emotional support from and within their organization which provides potential barriers to their satisfaction, or professional growth. Overall, the findings highlight that while teachers maintain positive professional self-conceptions and value their societal contributions, there is a need to strengthen recognition, support, and growth opportunities within institutions to consolidate a stronger sense of professional identity.

**Sampling adequacy and suitability for factor analysis**

The Kaiser-Meyer-Olkin (KMO) measure of sampling adequacy was 0.796, which falls in the acceptable range, indicating that the data were suitable for factor analysis. Bartlett's test of Sphericity was statistically significant ($\chi^2 = 1188.512$,

**Table 3. Item used for developing scale along mean item scores.**

| Sr. # | Items | Direction of Coding | Mean | Standard Deviation |
|---|---|---|---|---|
| 1 | Rewind Time | R | 2.61 | 1.248 |
| 2 | I feel proud to be a medical teacher | F | 1.58 | 0.724 |
| 3 | It syncs well with family life | F | 1.55 | 0.771 |
| 4 | My seniors encourage me grow professionally | F | 2.24 | 1.126 |
| 5 | I wish I could practice as a part time Physician | R | 3.50 | 1.245 |
| 6 | My hard work is usually acknowledged at workplace | F | 2.35 | 1.053 |
| 7 | I have started accepting people the way they | F | 4.06 | 0.735 |
| 8 | I stay calm in stressful situations | F | 2.23 | 0.790 |
| 9 | I take my students' words as personal evaluation of myself | F | 2.06 | 0.856 |
| 10 | I keep myself updated with latest trends in my field | F | 1.85 | 0.647 |
| 11 | I foresee myself in a better position in my career growth | F | 2.01 | 0.802 |
| 12 | I am glad I bring positive changes in society | F | 1.87 | 0.738 |
| 13 | The only reward I get at workplace is, more work. | R | 3.46 | 1.170 |
| 14 | I am positively challenged at workplace | F | 2.34 | 0.919 |
| 15 | Teaching comes naturally to me | F | 1.94 | 0.898 |
| 16 | I feel accomplished when I see my students progressing | F | 1.36 | 0.539 |
| 17 | People think medical teachers are not doctors | R | 3.80 | 1.089 |
| 18 | I consider my profession as a legacy (sadqaejaariah) | F | 1.69 | 0.801 |
| 19 | I enjoy liberty to exercise innovative ideas at workplace | F | 2.10 | 0.976 |
| 20 | I frequently attend workshops, conferences and seminars etc | F | 2.05 | 0.910 |

df = 190, p < 0.001), confirming that the correlation matrix was not an identity matrix and that sufficient inter-item correlations existed to proceed with factor analysis.

The results of factor analysis are shown in **Table 4** and **Table 5**. We considered those items who have correlation above 0.30. The factor loading for each construct indicates how strongly each item correlates with its respective factor. This factor analysis explored the presence of 5 factors with eigenvalues exceeding 1 (**See** Fig 1). Exploratory factor analysis of these items using the scores of 301 respondents indicated a five-factor structure. The initial eigenvalues for factors 1–5 were 4.308, 1.843, 1.575, 1.231, and 1.138 respectively. The percentages of variance for factors 1–5 were 21.54, 9.21, 7.87, 6.16, and 5.69%, respectively. All items had a component coefficient over 0.3, and the cumulative percentage of all five factors was 50.47%. Cronbach's alpha of the 20 items was 0.648which is approximately acceptable value of Cronbach's alpha.

Promax rotated pattern matrix coefficients are shown in Table 4. Items demonstrated meaningful loadings on their respective components, indicating that the scale successfully captured distinct yet related dimensions of medical teachers' professional attitudes, motivations, and behaviors. The pattern matrix coefficients showed that items clustered logically, with minimal cross-loadings of concern, supporting both the statistical and theoretical integrity of the factor solution. Communalities ranged from low to moderate, suggesting that each item contributed uniquely to the latent structure while still sharing adequate variance with the extracted factors. Collectively, the emergent factors reflect integrated constructs that align with the conceptual framework of the scale and demonstrate internal consistency in measuring the targeted domains.

Table 5 presents the extracted factors along with their corresponding item loadings, demonstrating a coherent and theoretically aligned factor structure. The names of the five factors were as follows: factor 1: Community of practice, factor 2: Legacy, factor 3: Job satisfaction, factor 4: Commitment to excellence and factor 5:Resilience. The items grouped under each factor show sufficiently strong and conceptually consistent loadings, indicating that respondents' perceptions

**Table 4. Pattern matrix components of the five factors of the developing scale after factor analysis using Promax rotation with Kaiser normalization.**

| Items | Factor loading | | | | | Communalities extraction |
|---|---|---|---|---|---|---|
| | 1 | 2 | 3 | 4 | 5 | |
| I wish I could rewind the time and chose clinical field | −.152 | −.042 | .146 | **.804** | −.015 | .378 |
| I feel proud to be a medical teacher | .177 | .364 | −.133 | **−.486** | .142 | .329 |
| It syncs well with family life | **.585** | .168 | −.167 | −.051 | .006 | .208 |
| My seniors encourage me grow professionally | **.736** | −.095 | .186 | .100 | .078 | .436 |
| I wish I could practice as a part time physician | .102 | −.056 | .026 | **.753** | .084 | .251 |
| My hard work is usually acknowledged at workplace | **.713** | −.106 | .179 | −.027 | .072 | .430 |
| I have started accepting people the way they are | −.143 | .021 | .225 | −.047 | **−.854** | .165 |
| I stay calm in stressful situations | −.138 | .003 | .320 | −.024 | **.607** | .159 |
| I take my students' words as personal evaluation of myself | −.212 | **.562** | −.029 | .283 | .202 | .166 |
| I keep myself updated with latest trends in my field | .035 | .262 | **.621** | .080 | −.129 | .292 |
| I foresee myself in a better position in my career growth | .263 | .247 | **.333** | .042 | −.058 | .250 |
| I am glad I bring positive change in society | .238 | **.595** | .167 | −.004 | .020 | .469 |
| The only reward I get at workplace is, more work | **−.592** | .314 | .017 | .126 | .125 | .158 |
| I am positively challenged at workplace | .440 | −.009 | **.447** | .095 | −.009 | .336 |
| Teaching comes naturally to me | −.018 | **.670** | .101 | −.123 | −.296 | .274 |
| I feel accomplished when I see my students progressing | −.059 | **.514** | .142 | −.248 | .098 | .282 |
| People think medical teachers are not doctors | .240 | .242 | −.529 | **.400** | −.093 | .094 |
| I consider my work as a legacy(sadqaejaariah) | −.193 | **.731** | .092 | −.040 | .027 | .298 |
| I enjoy liberty to exercise innovative ideas at workplace | .278 | .011 | **.419** | −.026 | .153 | .267 |
| I frequently attend workshops, conferences and seminars etc. | .055 | .150 | **.613** | .156 | −.044 | .253 |

clustered meaningfully around the intended constructs. Items related to workplace support, recognition, purpose, professional growth, job-related attitudes, and personal regulation loaded appropriately onto their respective components, reflecting clear differentiation among the latent dimensions. The magnitude and direction of factor loadings suggest that each item contributes uniquely to its factor while maintaining conceptual relevance to the broader domain it represents. The presence of both high and moderate loadings indicates a balanced distribution of item contributions across factors, while the limited negative loadings fall within acceptable interpretive boundaries and do not compromise the structure. Overall, these results provide evidence that the extracted factors are well-defined, internally coherent, and empirically representative of the constructs the scale aims to measure.

I.  **Community of practice:** First factor, Community of practice range from 0.447 to 0.736, indicates high positive loadings, while one item has negative loading. These higher values show that medical teachers who experience strong professional support such as encouragement from seniors and acknowledgment of their hard work, tend to have a more positive perception of their work environment. However, those who feel that their efforts only result in more workloads may develop dissatisfaction. At the same time, many teachers find personal fulfillment in their role, viewing teaching to create a lasting impact.

II.  **Legacy (Sadqaejaariah):** Sadqaejaariah is an Islamic concept of a type of deed which continues to earn rewards even after death of that individual, because the charity multiplies and continues to benefit other people long-term, even in upcoming generations. This can be better explained in the medical educational context as imparting knowledge and skill to a medical student will benefit not only one person, but his family, patients and society. Therefore, the reward of a teacher multiplies when his student spreads this knowledge and skill to another person and so on. This

**Table 5. Factors extracted on the basis of items.**

| Factors with items | Factor loadings |
| --- | --- |
| **Community of Practice** | |
| It syncs well with family life | 0.585 |
| My seniors encourage me grow professionally | 0.736 |
| My hard work is usually acknowledged at workplace | 0.713 |
| The only reward I get at workplace is, more work | 0.592 |
| I am positively challenged at workplace | 0.447 |
| **Legacy** | |
| I take my students' words as personal evaluation of myself | 0.562 |
| I am glad I bring positive changes in society | 0.595 |
| Teaching comes naturally to me | 0.670 |
| I feel accomplished when I see my students progressing | 0.514 |
| I consider my work as a legacy (sadqa e jaariah) | 0.731 |
| **job Satisfaction** | |
| I wish I could rewind the time and chose clinical field | 0.804 |
| I feel proud to be a medical teacher | −0.486 |
| I wish I could practice as a part-time physician | 0.753 |
| **Commitment to Excellence** | |
| I keep myself updated with the latest trends in my field | 0.621 |
| I foresee myself in a better position in my career growth | 0.333 |
| People think medical teachers are not doctors | −0.529 |
| I enjoy liberty to exercise innovative ideas at workplace | 0.419 |
| I frequently attend workshops, conferences, seminars etc. | 0.613 |
| **Resilience** | |
| I have started accepting people the way they are | 0.854 |
| I stay calm in stressful situations | 0.320 |

factor reflects the intrinsic motivation, personal fulfillment and the perception of teaching as a meaningful profession which ranges from 0.514 to 0.731. The belief that their work contributes positively to society and serves as a legacy (Sadqa e jaariah) motivates them to stay engaged.

III. **Job Satisfaction:** Third component shows some conflicting emotions regarding career choices and professional fulfillment. The strong loading with values 0.804 reflects that many medical teachers have lingering regrets about not choosing a clinical career. Similarly, an item having strength of 0.753 demonstrates that some teachers still have an interest in clinical work, while those who regret their career choice are less likely to take pride in their teaching role having a negative correlation of strength −0.486.

IV. **Commitment to excellence:** Fourth factor ranges from 0.333 to 0.621 and demonstrates that, however, teachers may feel underappreciated by society (factor loading of item is −0.529), their strong commitment to excellence and professional growth help them survive. They exercise innovative ideas at workplace, participate in continuous professional development opportunities and are optimistic for their future endeavors.

V. **Resilience:** Fifth factor demonstrates high and positive loadings (0.320 to 0.854) which reflects that medical teachers are likely to be emotionally resilient and capable of managing workplace challenges with a balanced mindset.

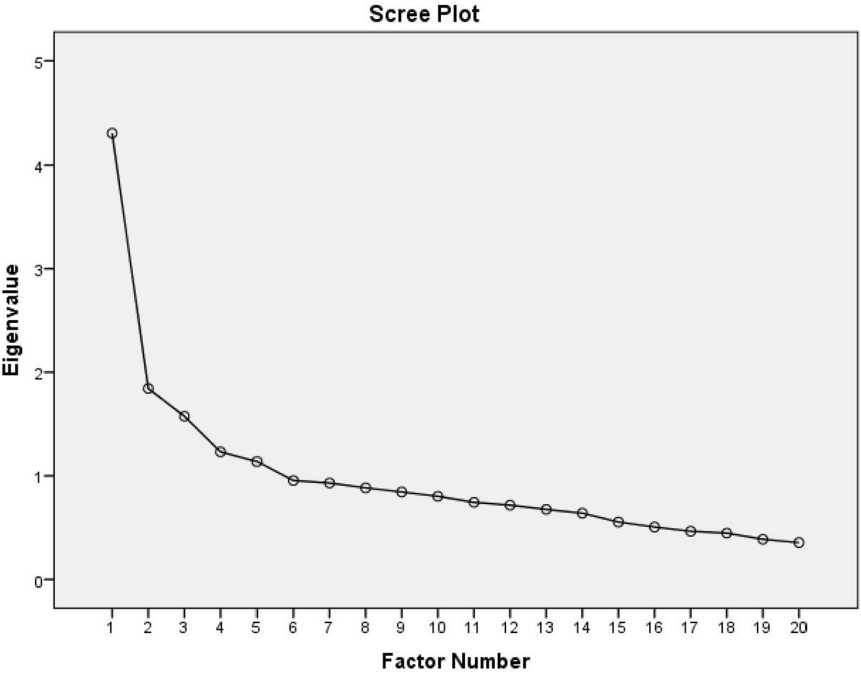

**Fig 1. Scree plot.**

## Discussion

This study aimed to explore factors required to form professional identity of basic sciences medical teachers. The analysis reveals that the smooth fusion of a teacher's role into their identity is either facilitated or hampered by behavior and messages from the community of practice, society and students.

Cruess et al in 2015 stated that existing identities and personal values when augmented by socialization process, initiate the identity formation process [25]. This statement builds up our study results which coincide with the results of a recent study of PIF in a non-western setting in 2021. It states that passion, religious beliefs, and societal recognition influence motivation of medical teachers. In a non-western culture, female medical teachers deliberately prioritize family responsibilities when choosing their career path and teaching is perceived as an opportunity to practice good deeds, daily [26]. This factor has been described by us as Legacy as a strong contributor to identity formation of medical teachers. In our context, both male and female teachers chose this career path for legacy and better work-life balance (See item 3 and 18 in Table 1). However, societal apathy, rather than societal recognition, is reported, more by female teachers (See item 17 in Table 1). This is reinforced by another study, in 2018, on experiences of surgical residents where female residents report disregard by patients and physicians more frequent than male residents. They face aggressive behaviors from attending and support staff, more often than males [27]. The same study also states that females have more concerns over future career advancements [27], which is also reflected in our study results where male teachers see better prospects than female teachers (See item 11 in Table 1). A recent study in Pakistan has also highlighted this gender bias. It reports that female faculty face challenges in achieving a stable work-life balance, and is under-represented in leadership roles in medical and dental colleges [28]. Another study in USA in 2022 has also showcased that cultural expectations of women manifest as microaggressions and stereotype threats and result in inequities and implicit biases for female physicians [29]. The study also reported that females experience work life imbalance.

Similar to the findings of our study, religious values and faith in God is also identified as one of the constructs of medical professionalism, in Arab countries [30]. Our study results align with recent 2023 US research on basic sciences medical teachers, identifying job satisfaction, rewards, and communities of practice as key factors for retaining and motivating teachers to excel [3].

Studies show that personal and contextual resources are essential for teachers to stay in their profession, overcome challenges, and succeed. Globally, it is agreed that personal resources like motivation and social competence, along with coping strategies such as problem-solving and work-life balance, help teachers build resilience [30].

Our study results align with a 2017 study conducted in the Netherlands, which found that new entrants into undergraduate medical education face challenges in forming their professional identity due to feelings of marginalization and low self-esteem compared to their clinical peers. These teachers often view teaching as a low-status occupation, perceived as less valuable than clinical work or research. Additionally, pursuing a career in medical education is frequently not a first choice and is often not well planned. Consequently, teaching roles are adopted through collegial support, leading to a gradual identification with the role of an educator [31].

Our study results presented in Table 1, show that in initial years of their career, teachers show dissatisfaction and desire to choose the clinical field. This is because of their compartmentalized (identity is explicit and separate from clinicians) and hierarchical identity (identity is positioned as inferior to clinicians) [5]. During the early stages of their careers, BSMTs often face challenges and, therefore, struggle in establishing their professional identity. However, as they advance to senior positions or gain authority, they achieve identity integration. As shown in Table 2, after accumulating 15 or more years of experience and attaining higher academic titles or positions, BSMTs develop a coherent professional identity [5]. Cantillon et al states that the identity of medical teachers is formed and reinforced through their experiences and trials, continuous reflection on these experiences, and by emulating or dismissing role models [5].

Experienced teachers generally consider teaching as an integral part of their professional identity. Most of them consider it as both a professional responsibility and a moral commitment to give back to their field and society at large. Additionally, they have a strong desire to contribute to the development of future generations of professionals. Teachers often express satisfaction and joy in the process of teaching itself, finding fulfillment in witnessing their students' progress and taking pride in their achievements [31].

Faculty development programs for identity development of basic sciences faculty must be designed emphasizing (GRIT) Growth, Resilience, Inspiration, and Tenacity [29]. This can be achieved by investing in formal training programs, provision of clear career trajectories, revising promotion and tenure criteria, faculty incentives, positive administrative role [31]. We agree with existing literature emphasizing the need to ensure that basic health sciences faculty feel a sense of belonging and being valued [31]. This can be fostered by arranging ceremonial celebrations for faculty where their contributions are appreciated, for example by awards, rewards, praise by students or public speakers, hearing positive stories about teaching and its role in societal modeling etc. Such events and activities can also provide beginning teachers a platform to find inspirations, role models and mentors [31]. Moreover, gender discrimination can be countered by fostering a supportive community of female medical teachers through collegial support groups, garnering support from male colleagues, encouraging personal and professional development (i.e., self-advocacy, negotiation skills, and mentorship opportunities) and systemwide policy changes that promote equity and empower women to assume leadership roles [29].

The strengths of our study are that participants from varied designations, qualifications and years of service were sampled. Though more females participated in our study, it is because only fewer males chose this profession. The reasons are discussed in our earlier qualitative study [9]. In general, no significant difference in their opinions was found.

## Limitations

All respondents were in Pakistan, so the findings may be applicable to our local context. Although the sample size was 301, it cannot be concluded that those who did not participate in the study will have similar experiences to those who have

responded to the questionnaire. There is a possibility that some of the items may not be appropriate in other cultures as cultural or systemic differences across countries may affect the applicability of the results.

Furthermore, little is known about the identity characteristics of BSMTs, thus it cannot be ascertained to what extent the data reflects the diversity of BSMT, in general. Future studies on BSMTs in other contexts are needed to explore perspectives of diverse BSMTs on PIF. One of the limitations of this study is that the psychometric validation of the instrument was limited to exploratory factor analysis (EFA). While EFA is appropriate for identifying the underlying factor structure during the initial development phase, the absence of confirmatory factor analysis (CFA) means that the proposed factor structure has not yet been validated in an independent sample. Future research should include CFA to confirm the factor structure, assess model fit, and further establish the construct validity of the instrument.

## Conclusion

This is the first report to develop a scale that quantitatively evaluates professional identity formation in basic sciences medical teachers. The developing questionnaire has a five-factor structure and is reliable. Hence, the Community of Practice is one major factor, Legacy is second factor, job Satisfaction is third, Commitment to Excellence is fourth, and Resilience is the fifth main factor, proposing that CLaSiC-R is the identity blueprint for professional identity of basic sciences medical teachers.

The CLaSiC-R model provides a structured understanding of the key psychological and contextual drivers that shape the professional identity of medical teachers in the basic health sciences. Each factor is interdependent, reflecting the holistic and evolving nature of professional identity formation. Recognizing and supporting these dimensions—particularly through faculty development programs, institutional recognition, and mentoring—can foster a stronger, more resilient, and satisfied academic workforce.

It can form the foundation for structured faculty development programs. Policies can mandate mentorship programs to strengthen community of practice and legacy, continuous training in emotional intelligence, communication, and interpersonal skills to promote social and emotional competence.

We encourage medical schools to embed the CLaSIC-R model into their faculty handbooks or mission for teachers. Moreover, institutions must promote a culture that values legacy and contribution and discourage sabotaging or discouraging attitude of senior peers. Society must be made aware of the significant role basic medical sciences teachers play as foundational to future clinicians and researchers. Institutions must invest in wellness initiatives and stress management of teachers to promote resilience. In addition, there must be career advancement pathways, respectable salary and a reward/recognition system in place for their job satisfaction.

Furthermore, this model offers a culturally grounded framework for guiding future research and policies within medical education in Pakistan and similar contexts. By identifying the components that matter most to teachers, stakeholders can align institutional strategies to nurture professional identity, thereby improving teaching quality and educational outcomes.

## Supporting information

**S1 Appendix. Complete questionnaire used for data collection.**
(PDF)

**S1 Data. Dataset used for statistical analyses.**
(CSV)

## Author contributions

**Conceptualization:** Faiza Kiran.

**Data curation:** Shazia Irum, Samreen Misbah, Asiya Zahoor.

**Formal analysis:** Nadia Shabnam.

**Investigation:** Rukhsana Ayub.

**Methodology:** Faiza Kiran, Nadia Shabnam, Asiya Zahoor, Rukhsana Ayub.

**Project administration:** Faiza Kiran, Asiya Zahoor.

**Resources:** Asiya Zahoor.

**Software:** Nadia Shabnam.

**Supervision:** Faiza Kiran, Asiya Zahoor, Rukhsana Ayub.

**Validation:** Nadia Shabnam.

**Writing – original draft:** Faiza Kiran.

**Writing – review & editing:** Nadia Shabnam, Shazia Irum, Samreen Misbah.

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
