## [Decision Letter · Decision Letter 0]

11 Jun 2025

Dear Dr. Kiran,

Thank you for submitting your manuscript to PLOS ONE. After careful consideration, we feel that it has merit but does not fully meet PLOS ONE’s publication criteria as it currently stands. Therefore, we invite you to submit a revised version of the manuscript that addresses the points raised during the review process.

Comments by Reviewer 1 are quite pertinent and need to be addressed adequately. Bothe reviewers have note lack of Confirmatory Factor Analysis (CFA) in the study which augment the need of its inclusion.

We look forward to receiving your revised manuscript.

Kind regards,

Aamir Ijaz, MD, FCPS, FRCP, MCPS-HPE

Academic Editor

PLOS ONE

Journal Requirements:

4. Please include captions for your Supporting Information files at the end of your manuscript, and update any in-text citations to match accordingly. Please see our Supporting Information guidelines for more information: http://journals.plos.org/plosone/s/supporting-information .

Additional Editor Comments:

Comments by Reviewer 1 are quite pertinent and need to be addressed adequately. Bothe reviewers have note lack of Confirmatory Factor Analysis (CFA) in the study which augment the need of its inclusion.

Reviewers' comments:

Reviewer's Responses to Questions

**Comments to the Author**

1. Is the manuscript technically sound, and do the data support the conclusions?

Reviewer #1: Partly

Reviewer #2: Partly

2. Has the statistical analysis been performed appropriately and rigorously?

Reviewer #1: No

Reviewer #2: Yes

3. Have the authors made all data underlying the findings in their manuscript fully available?

Reviewer #1: No

Reviewer #2: No

4. Is the manuscript presented in an intelligible fashion and written in standard English?

Reviewer #1: Yes

Reviewer #2: Yes

Reviewer #1: 1. Lack of Justification for Sample Size

Location: Page 4 – Sampling and Participants

Excerpt: "A total of 301 educators teaching basic medical sciences provided their perceptions..."

Comment: The paper does not explain the methodological justification for the sample size (e.g., based on power analysis or factor analysis requirements), which is critical in survey-based studies.

2. No Confirmatory Factor Analysis (CFA) to Validate the Construct

Location: Page 6 – Phase 3 – Scale Evaluation

Excerpt: “...factor structure of the instrument, using exploratory factor analysis...”

Comment: The study relies only on EFA and does not follow up with CFA to confirm the factor structure. This weakens the psychometric validation process of the proposed instrument.

3. Low Explained Variance from Factor Model

Location: Page 2 (Abstract) and Page 7

Excerpt: “...five variables of EFA, which explained 21% of the variance.”

Comment: An explained variance of only 21% is relatively low for a scale development study and should either be justified or presented as a limitation.

4. Unusual Likert Scale Direction Not Clarified

Location: Page 5

Excerpt: “Likert scale ranging from 1: strongly agree to 5: strongly disagree.”

Comment: The use of a reverse Likert scale (where 1 = agree and 5 = disagree) deviates from standard practice and should be explained explicitly to prevent confusion in interpretation and scoring.

5. Gender-Based Differences Not Fully Discussed

Location: Page 7 and Table 1

Excerpt: “...more females 179(78%) seem affected by societal apathy...”

Comment: The gender dimension is reported but not critically explored in the discussion section. A more thorough analysis of gender-specific professional identity challenges would strengthen the paper.

6. Limited Discussion on Generalizability

Location: Pages 10–11 – Limitations

Excerpt: “...participants were recruited participants from our own country only...”

Comment: The limitations section mentions geographical scope but fails to address how cultural or systemic differences across countries may affect the applicability of the results.

7. No Control for Response Bias in Online Data Collection

Location: Pages 5–6 – Survey Distribution

Excerpt: “...distributed on social media platforms (WhatsApp, email, Facebook)...”

Comment: The authors do not mention how they controlled for response bias or social desirability bias, which are important in open online surveys.

8. Lack of Comparison with Existing Professional Identity Scales

Location: Entire Discussion section

Excerpt: No discussion found on prior instruments such as PIQ, DIT, or PIF.

Comment: The study does not compare its findings or constructs with established scales on professional identity, missing an opportunity to highlight the uniqueness or alignment of its proposed tool.

9. Absence of Graphical Data Presentation

Location: Pages 7–9 – Results

Excerpt: Results only presented in tables (e.g., Table 1, Table 2, Table 3)

Comment: Visual aids such as scree plots, factor loading diagrams, or conceptual maps would enhance clarity and reader engagement.

10. Use of Local Cultural Terms Without Explanation

Location: Table 1, Item 18

Excerpt: “I consider my profession as a legacy (sadqaejaariah)”

Comment: The term sadqa jariah, while culturally meaningful, may not be understood by international readers. A clearer academic or secular translation is necessary for broader accessibility.

Reviewer #2: The manuscript lacks a clearly articulated theoretical framework, and several claims are presented without appropriate referencing. The methodology appears to be an extension of a prior study conducted by the author; however, this connection should be more explicitly clarified. While the statistical methods employed for instrument validation are methodologically sound and robust, the presentation would benefit from more comprehensive supporting evidence. Additionally, incorporating Confirmatory Factor Analysis (CFA) could further strengthen the validity and generalizability of the instrument.

**Do you want your identity to be public for this peer review?** For information about this choice, including consent withdrawal, please see our Privacy Policy

Reviewer #1: **Yes:** MICHAEL CHRISTIAN

Reviewer #2: No

---

## [Author Response · Author response to Decision Letter 1]

11 Dec 2025

Firstly, a thanks to the reviewer for providing us such valuable suggestions to improve our manuscript.

Reviewer # 1:

The paper does not explain the methodological justification for the sample size (e.g., based on power analysis or factor analysis requirements), which is critical in survey-based studies.

Response:

Thank you for pointing this out. We acknowledge the importance of providing a methodological justification for the sample size in survey-based research. While a formal power analysis was not conducted prior to data collection, the sample size was determined based on established recommendations for factor analysis, which suggest at least10 participants per item. Given that our survey consisted of 20 items, the final sample size of participants was considered adequate for reliable analysis. This has now been clarified in the revised manuscript.

The study relies only on EFA and does not follow up with CFA to confirm the factor structure. This weakens the psychometric validation process of the proposed instrument.

Response:

Thank you for your valuable observation. We acknowledge that confirmatory factor analysis (CFA) is an important step in validating the factor structure identified through exploratory factor analysis (EFA). However, given that this study aimed to develop and preliminarily explore the underlying structure of the instrument, we focused on EFA as an appropriate first step in the psychometric validation process.

We agree that CFA would strengthen the validation of the instrument, and we have now clarified in the manuscript that future studies involving independent samples are planned to conduct CFA and further assess model fit, convergent validity, and discriminant validity. This limitation has been acknowledged and discussed in the revised “Limitations” section of the manuscript.

An explained variance of only 21% is relatively low for a scale development study and should either be justified or presented as a limitation.

Response:

Thank you for your observation. We apologize for the confusion—upon rechecking the analysis, the cumulative variance explained by the extracted factors is 50%, not 21% as previously noted. This level of explained variance is generally considered acceptable in the context of social and behavioral sciences, particularly in exploratory factor analysis.

The use of a reverse Likert scale (where 1 = agree and 5 = disagree) deviates from standard practice and should be explained explicitly to prevent confusion in interpretation and scoring.

Response:

Thank you for your observation. We apologize for the error-it has been corrected in revised manuscript.

The gender dimension is reported but not critically explored in the discussion section. A more thorough analysis of gender-specific professional identity challenges would strengthen the paper.

Response:

Thank you for your valuable suggestion. Gender specific challenges have been explored in literature and been added in the discussion section of the manuscript.

In a non-western culture, female medical teachers deliberately prioritize family responsibilities when choosing their career path and teaching is perceived as an opportunity to practice good deeds, daily(26). This factor has been described by us as Legacy as a strong contributor to identity formation of medical teachers. In our context, both male and female teachers chose this career path for legacy and better work-life balance; See item 3 and 18 in Table 1). However, societal apathy, rather than societal recognition, is reported, more by female teachers (See item 17 in Table 1). This is reinforced by another study, in 2018, on experiences of surgical residents where female residents report disregard by patients and physicians more frequent than male residents. They face aggressive behaviours from attending and support staff, more often than males(27). The same study also states that females have more concerns over future career advancements (27), which is also reflected in our study results where male teachers see better prospects than female teachers ( See item 11 in Table 1). A recent study in Pakistan has also highlighted this gender bias. It reports that female faculty face challenges in achieving a stable work-life balance, and is under-represented in leadership roles in medical and dental colleges(28). Another study in USA in 2022 has also showcased that cultural expectations of women manifest as microaggressions and stereotype threats and result in inequities and implicit biases for female physicians(29). The study also reported that females experience work life imbalance.

The limitations section mentions geographical scope but fails to address how cultural or systemic differences across countries may affect the applicability of the results.

Response:

All respondents were in Pakistan, so the findings may be applicable to our local context. Although the sample size was 301, it cannot be concluded that those who did not participate in the study will have similar experiences to those who have responded to the questionnaire. There is a possibility that some of the items may not be appropriate in other cultures as cultural or systemic differences across countries may affect the applicability of the results.

The authors do not mention how they controlled response bias or social desirability bias, which are important in open online surveys.

Response:

Thank you for this important observation. We agree that response bias and social desirability bias are potential concerns in self-reported, open online surveys. We have now added a description of these measures in the revised manuscript in methodology; under the phase 2 and sampling section as well as in ethical considerations. Additionally, we have acknowledged the potential for residual response bias as a limitation of the study.

In our study, we took several steps to minimize these biases. First, the survey was administered anonymously, and participants were assured that their responses would remain confidential and be used solely for research purposes. Second, no identifying information was collected, which may have helped reduce the tendency to provide socially desirable responses.

The survey was designed as a self-administered and self-completed online questionnaire. As compared to face-to-face or phone methods, distributing scales online removes interviewers, and ensures anonymity and confidentiality. Responses were taken on a Likert scale which makes it more feasible for the respondents to answer. In addition, randomized response format was ensured by using indirect or reverse questioning, which tend to reduce common methods and social desirability bias(18).

The study does not compare its findings or constructs with established scales on professional identity, missing an opportunity to highlight the uniqueness or alignment of its proposed tool.

Response:

Thank you for the suggestion. It has been explored and added in the introduction section.

Previous studies have extracted factors affecting professional identity of medical students, trainees, physicians and basic sciences medical teachers, by qualitative analysis. Only fewer studies have approached the problem with quantifiable means; an essential measure to quantify attitudes, values and beliefs of physicians, trainees, and medical teachers. Of these a 10-item Professional identity questionnaire (PIQ) (12) and a 9-item Professional Self Identity Questionnaire (PSIQ) (13) in 2021, were developed for medical students. Another one, in 2015, was developed to quantify identity transformation in medical students, starting from premedical era till graduation. It has 10 key aspects, 06 domains and 30 subdomains(14). Only one questionnaire by Tagawa in 2019, a 15-item scale, was found which measures PIF in residents, experienced instructors and medical students (15).

In 2017, a review on teacher identity, in the university context, identified 59 studies; 57 of whom applied qualitative methods, such as interviews or focus group discussions to assess teacher identity. Qualitative assessments of teacher identity, though gives broader scope, but fails to provide an economical application to repeated measurements or large-scale assessments(16), for example when questioning medical teachers in entire hospital or medical college. In 2006, an instrument measuring physicians’ teacher identity was developed by Starr et al., including 37 items nested in nine scales designed from previously conducted interviews (17).

Though an important area, not a single study has been found where professional identity of basic sciences medical teachers has been measured by quantitative means. Understanding the specific factors that shape the PIF of BSMT is crucial not only for supporting individual faculty development but also for improving the overall quality of medical education. A deeper insight into this process can inform institutional policies, faculty development programs, and mentorship models tailored to local sociocultural realities.

Visual aids such as scree plots, factor loading diagrams, or conceptual maps would enhance clarity and reader engagement.

Response:

Thank you for this suggestion, we have added the scree plot in the revised manuscript.

The term sadqa jariah, while culturally meaningful, may not be understood by international readers. A clearer academic or secular translation is necessary for broader accessibility.

Response:

Thank you for pointing this out. We have explained the term in phase 3 of scale evaluation.

Sadqaejaariah is an Islamic concept of a type of deed which continues to earn rewards even after death of individual, because the charity multiplies and continues to benefit people long-term, even in upcoming generations. This can be better explained in the medical educational context as imparting knowledge and skill to a medical student or doctor will benefit him, his family, the people he influences such as patients and society. The reward multiplies when he spreads this knowledge and skill to another person and so on.

Reviewer # 2

The manuscript lacks a clearly articulated theoretical framework, and several claims are presented without appropriate referencing. The methodology appears to be an extension of a prior study conducted by the author; however, this connection should be more explicitly clarified. While the statistical methods employed for instrument validation are methodologically sound and robust, the presentation would benefit from more comprehensive supporting evidence. Additionally, incorporating Confirmatory Factor Analysis (CFA) could further strengthen the validity and generalizability of the instrument.

Thank you for your positive feedback regarding the statistical methods used. We appreciate your suggestion to enhance the presentation with more comprehensive supporting evidence. In the revised manuscript, we have elaborated on the interpretation of the exploratory factor analysis (EFA) results, including factor loadings, eigenvalues, and item retention criteria, to provide clearer justification for the instrument's structure.

Regarding Confirmatory Factor Analysis (CFA), we agree that its inclusion would strengthen psychometric validation. However, as this study represents the initial phase of instrument development, our focus was on EFA to explore the underlying factor structure. We have now explicitly mentioned this in the manuscript and noted the need for CFA in future studies to confirm and generalize the factor structure in independent samples. This point has also been added to the Limitations section.

---

## [Editor Report · Decision Letter 1]

18 Dec 2025

The Identity Blueprint: Decoding the Professional Identity of Basic Sciences Medical Teachers in Pakistan by developing and pilot testing a questionnaire

PONE-D-25-20758R1

Dear Dr. Kiran,

We’re pleased to inform you that your manuscript has been judged scientifically suitable for publication and will be formally accepted for publication once it meets all outstanding technical requirements.

Kind regards,

Aamir Ijaz, MD, FCPS, FRCP, MCPS-HPE

Academic Editor

PLOS One
---

## [Editor Report · Acceptance letter]

PONE-D-25-20758R1

PLOS One

Dear Dr. Kiran,

I'm pleased to inform you that your manuscript has been deemed suitable for publication in PLOS One. Congratulations! Your manuscript is now being handed over to our production team.

Kind regards,

on behalf of

Professor Aamir Ijaz

Academic Editor

PLOS One